# The CPD Data Set: Personnel, Use of Force, and Complaints in the Chicago Police Department

**Thibaut Horel**
LIDS, MIT
thibauth@mit.edu

**Lorenzo Masoero**
CSAIL, MIT
lom@mit.edu

**Raj Agrawal**
ArbiLex
r.agrawal@csail.mit.edu

**Daria Roithmayr**
Gould School of Law
University of Southern California
droithmayr@law.usc.edu

**Trevor Campbell**
Department of Statistics
University of British Columbia
trevor@stat.ubc.ca

## Abstract

The lack of accessibility to data on policing has severely limited researchers' ability to conduct thorough quantitative analyses on police activity and behavior, particularly with regard to predicting and explaining police violence. In the present work, we provide a new dataset that contains information on the personnel, activities, use of force, and complaints in the Chicago Police Department (CPD). The raw data, obtained from the CPD via a series of requests under the Freedom of Information Act (FOIA), consists of 35 unlinked, inconsistent, and undocumented spreadsheets. Our paper provides a cleaned, linked, and documented version of this data that can be reproducibly generated via open source code. We provide a detailed description of the dataset contents, the procedures for cleaning the data, and summary statistics. The data have a rich variety of uses, such as prediction (e.g., predicting misconduct from officer traits, experience, and assigned units), network analysis (e.g., detecting communities within the social network of officers co-listed on complaints), spatiotemporal data analysis (e.g., investigating patterns of officer shooting events), causal inference (e.g., tracking the effects of new disciplinary practices, new training techniques, and new oversight on complaints and use of force), and much more. Access to this dataset will enable the machine learning community to meaningfully engage with the problem of police violence.

**Repository:** https://github.com/chicago-police-violence/data
**Dataset:** https://github.com/chicago-police-violence/data/releases/download/v0.1/cpd-dataset-v0.1.zip
**Instructions:** run `make` in the repository root directory to build the dataset in the `final/` folder from raw source files. See `README.md` for detailed instructions.

## 1 Introduction

Accurate and accessible data on policing is crucial to understanding and remedying the problem of police violence. Police misconduct inflicts harm on families and communities, particularly those of color in which police presence is pervasive [1]. In addition to inflicting harm, misconduct also undermines the foundation of trust between civilians and law enforcement, particularly when misconduct is not punished [2]. Accessible data enables researchers and policymakers to accurately determine causes, consequences, and potential remedies for police misconduct. However, data about

policing is severely limited, largely because state/local law enforcement agencies rarely disclose internally collected information [3]. Even when departments do disclose information, they often withhold data linked to individual officers and their professional relationships. As a result, scholars and policymakers often lack the data they need to accurately describe and remedy the problem of police misconduct. Rigorous quantitative analysis, which often depends on large amounts of data pertaining to individual officers, is particularly difficult.

In this work, we provide a dataset containing information on individual personnel, their activities, shootings, use of force, and complaints filed against them at the Chicago Police Department (CPD). Chicago is a particularly appropriate city for a case-study on policing, as it is a major metropolitan area with a large, diverse police force that has been investigated by the US Department of Justice due to its high rate of police-involved shootings [4]. We describe in detail the dataset contents, as well as our procedures for organizing, cleaning and linking the data across multiple source files in Section 3. We provide summary analyses of the available categories of data in Section 4. We finally propose a rich variety of uses for this dataset in Section 5.

The data on individual officers underlying our dataset comes from internal documents routinely generated by the CPD and disclosed per Freedom of Information Act requests. The CPD provided this data in thirty-five unlinked, inconsistent, error-prone, and undocumented spreadsheets. These spreadsheets were then published online and made available to researchers by the non-profit Invisible Institute [5]. We also draw from data on police shootings provided by the CPD's civilian oversight board. Prior to our work, linking information across the various heterogeneous data sources was a serious challenge: an officer described in one source is not uniquely identifiable as the same officer in other sources due to inconsistent fields, missing data, and time-varying attributes. The value of our work thus lies in (1) consolidating these heterogeneous files to create a clean, well-organized, and well-integrated dataset on individual officers, (2) providing code that reproducibly builds the dataset from the raw source files, and (3) a detailed description of the method we developed to carefully link data files.

## 2    Related Work

Existing public datasets on policing are extremely limited; few agencies disclose internal personnel data to the public [3], and other data collected by police (e.g., to build "early intervention" warning systems) remain confidential, even when scholars publish the results of their work on such systems [6]. In response to public pressure, however, a number of public datasets on policing have recently emerged: the Police Data Initiative, a collection of datasets from more than 130 state and local law enforcement agencies [7]; Stanford University's Open Policing Initiative, which contains data on traffic stops from around the country [8]; and the NYPD Misconduct Complaint Database, which includes complaints regarding officers in the New York Police Department [9]. However, these datasets do not include information about individual officer behavior, contain only raw, inaccessible data, or both. Furthermore, the Department of Justice (DOJ) only began last year to collect data on arrest-related deaths through the implementation of the "Death in Custody Reporting Act" of 2013[1] [10]. Under this act, states are required to provide data regarding the death of any person related to policing activity. But states often cannot compel local law enforcement to disclose the data in the absence of state legislation. Likewise, the Federal Bureau of Investigation (FBI) "National Use-of-Force Data Collection" program is voluntary [11].

Other work has drawn on the same raw data that we clean and integrate in this work. The Invisible Institute itself released a processed version of the data that has been used in previous work [12, 13]. However, the code underlying this release suffers from gaps in its methodology and documentation, which severely limits the usability and reproducibility of the data. The raw data has also been used directly in concert with litigation settlements involving CPD officers [14], but no public information is available on how the raw data were cleaned or linked.

---

[1]Efforts to implement and enforce the law have been delayed for several years. According to a 2018 report by the DOJ's Office of the Inspector General, this delay was due in part to confusion over the law's requirements and the failure of the DOJ to agree on a proposal for data collection.

# 3 The CPD Data

The original raw data released by the CPD, as well as the code to generate both the cleaned data and this document, are available in a GitHub repository at `https://github.com/chicago-police-violence/data`. This repository will serve as a long-term home for this data, its current and future releases, as well as discussions regarding improvements and extensions of the data processing code. Refer to README.md in the repository for details regarding system requirements and how to run the data processing code.

## 3.1 The raw data: origin, description, and challenges

**Origin.** The first raw data files in this repository were obtained by J. Kalven, an independent journalist, who filed Illinois Freedom of Information Act (FOIA) requests with the Chicago Police Department regarding complaints filed against officers. In Kalven v. City of Chicago [15], an Illinois appellate court issued a general ruling that documents bearing on allegations of police abuse are public information. Following the decision, the non-profit Invisible Institute began to collaborate with Kalven and the University of Chicago's Mandel Legal Aid Clinic to follow up on earlier FOIA requests and to file new ones. The data disclosed in response to these earlier and now ongoing FOIA requests were made available online as part of the Citizens Police Data Project [16]. These data form the basis of the cleaned and linked data set provided by the present work.

Table 1: Summary of the FOIA requests contained in our repository (blanks are missing entries).

| Request # | Received | Requested | Description | # Records |
|---|---|---|---|---|
| P0-58155 | 2017-04-17 | | Officer roster | 32 446 |
| P4-41436 | 2018-03-21 | | Officer roster | 14 634 |
| 16-1105 | 2016-03-11 | 2016-02-10 | Unit assignments | 114 630 |
| P0-52262 | 2016-12-04 | 2016-09-19 | Unit assignments | 115 987 |
| P0-46957 | 2016-06-29 | 2016-04-22 | Complaints (CPD) | 109 339 |
| 18-060-425 | 2018-08-28 | 2018-08-20 | Complaints (COPA) | 182 337 |
| P0-46360 | | | Tactical Response Reports | 67 019 |
| P0-46987 | 2016-05-13 | 2016-04-25 | Unit names | 237 |
| P0-61715 | | 2017-07-26 | Awards | 699 912 |
| P5-06887 | 2019-10-11 | 2019-07-19 | Awards | 60 556 |
| | 2017-09-27 | 2017-09-13 | Salary | 212 508 |

**Description.** The raw data files are contained in the `raw/` folder of the repository. Each subfolder corresponds to a FOIA request, which is generally identified by a request number. Table 1 gives an overview of all the requests that we include in our repository; this meta-information is also included in the `raw/datasets.csv` file in the repository. The subfolders contain the data provided by the city in response to their corresponding FOIA requests, which typically comprises multiple Excel spreadsheets. In addition, when available, the subfolders contain formal correspondence regarding the request, which often provides useful contextual information in understanding the data.

In particular, the raw data files contained in the repository provide the following information:

**Officer roster:** A list of all officers (past and present) employed by the CPD along with attributes such as year of birth, age, race, gender, appointment date, resignation date, etc.

**Unit assignments:** The CPD is organized into over 200 units. Each officer can be assigned to one or multiple units and these assignments can change over time. The unit assignment datasets contain one record for each officer and each unit they were assigned to, including the start date and end date of this assignment.

**Complaints:** Formal complaints against police officers, filed both by citizens and internally within the department. Complaints are identified by a complaint number. There is one record for each complaint and each officer listed on the complaint, indicating the allegation made against them, result of the investigation of the allegation (with possible sanction), etc.

**Tactical Response Reports (TRRs):** These are forms that officers are required to file after each incident for which the officer's response involved use of force. There is one record for each incident and each officer involved in the incident. Each record contains details about the incident (such as time and location), the officer involved and the subject of the use of force. In case one or multiple weapons were used, detailed information about each use is also provided including, for firearms, the number of discharges and the object struck at each discharge.

**Unit names:** The (human-readable) name of each past and present unit in the CPD. These names provide information about the function of each unit and also appear occasionally where unit numbers are listed in the other data files.

**Awards:** A list of all awards requested for officers in the CPD, including award tracking number, reference number, award type, request date, requester name, etc. According to a department directive [17], awards are given "in recognition of outstanding acts of bravery and heroism, outstanding accomplishments, and exceptional performance."

**Salary:** A list of officers including their salary, position, and pay grade.

**Challenges.** Despite the richness of the information contained in these FOIA data releases, using these releases to investigate the activities of the CPD is not straightforward. In particular, police officers are not uniquely identified across datasets; there is a priori no reliable way to know whether, for example, an officer listed on a Complaint is the same individual as an officer with similar attributes listed on a Tactical Response Report. Therefore, one is forced to link officers across datasets using a restricted set of attributes, which introduces the following challenges:

- *Time-varying attributes:* many attributes change over the course of an officer's career in the CPD, such as their unit assignments, rank, and badge number (referred to as a "star" in the data). Perhaps surprisingly, some attributes which would usually be considered stable and useful identifiers also change over time. For example, surnames change when officers marry, and appointment dates change when database entries are corrected internally.

- *Inconsistent entries:* various choices were made by the CPD to decide which officers to include in each dataset. There are, for example, officers missing from the roster or unit assignment data, but present in the salary data. Furthermore, the same attribute can appear under different names in different datasets and sometimes have ambiguous meanings: for example, the salary data contains two different attributes for the appointment date. Fields that are available or missing per record also vary across databases.

- *Duplicate entries:* probably due to internal errors in the CPD data infrastructure, some officers are sometimes duplicated in the roster and unit assignment data: they appear twice in the same dataset, as two different individuals but with the exact same attributes. One of the two "copies" of each duplicate officer is inactive and never appears in the rest of the data, but introduces ambiguities to uniquely identify officers across datasets.

- *Systematic errors:* an unusual difficulty arose from the unit assignment data, in which a significant fraction of the assignments have an end date chronologically preceding the start date. This appears to be a systematic error made during either data entry or the process of releasing the data per FOIA requests. A close inspection of the pattern of errors revealed that the faulty records cannot be fixed by simply swapping the start date with the end date.

- *Multiple internal sources:* the salary data comes from a different database. In particular, the officers' names appearing in the salary data do not match exactly the names from other datasets, which makes linking this data to the other datasets particularly challenging.

## 3.2  Data Cleaning and Linking

A major contribution of the present work is to address all the above challenges by carefully cleaning and linking the original datasets. The output of this process is a collection of comma-separated values (CSV) files corresponding to the different entities in the *Description* paragraph above. Importantly, each officer is uniquely identified by a hexadecimal string across output files. To produce the final, cleaned dataset, run `make` in the root of the repository. The output files will be found in the `final/` folder; please refer to `description.md` for a description of the content of each these files. We now give an overview of each processing steps; more details can be found in Appendix B.

**Initial Cleaning.** In the first step, we produce uniformly formatted CSV files from the raw Excel spreadsheets. The decisions made at this stage are straightforward and involve no subjective judgement; they consist of (i) unifying attribute names and values across datasets, (ii) parsing integers and dates into a standard format, and (iii) concatenating records of the same type when they were split over multiple spreadsheets. Running `make prepare` in the root of the repository performs these steps, with output that may be found in the `tidy/` folder. This cleaned—but not linked—version of the data might be useful in applications that require different choices to be made during linking.

**Linking and merging datasets.** Next, we address the challenge of uniquely identifying officers across datasets. At a high level, our procedure grows a population of uniquely identified officers by sequentially examining each FOIA release. Each unique officer in this population is assigned a unique identification (UID), which is a random hexadecimal string (e.g. `9bc51eef-c37b-4eff-a14d-7e69f56b3d1e`), and to each UID is associated a list of *officer profiles*. There is one *officer profile* for each unique officer and each FOIA release in which the officer appears. Each profile lists the identifying attributes of the officer as they appear in the given data release. In detail, for each FOIA release:

1. We build a list of all the *officer profiles* appearing in this release.

2. For each officer profile, we attempt to *match* it against the profiles of the population of unique officers constructed so far from previously examined FOIA releases.

   - If the match is successful, we have identified a unique officer in the population whose profiles unambiguously match with the current profile. In this case, we simply attach the current profile to this officer and UID, and the population does not grow.
   - If the match is unsuccessful, we add a new officer with a new UID to the population of unique officers and attach the current profile to this new officer.

The *match* operation in step 2. is achieved by an iterative pairwise procedure that we developed and whose details are given in Appendix B.2. After all FOIA releases have been processed, the population contains the set of all unique officers appearing in the original data. Each officer is represented by a UID and a collection of profiles, representing the various ways in which this unique officer appears across different datasets. These profiles can be found in the file `final/officer_profiles.csv`. At this point, the `final/` folder also contains one file for each type of record (complaints, tactical response reports, etc). Within these files, officers are identified solely by their UID and other attributes are removed; this avoids duplication of information since these attributes are redundant with those found in `final/officer_profiles.csv`.

**Final steps.** For convenience, we consolidate the different profiles of each officer into a single, canonical profile as follows. For each officer and each identifying attribute, we choose the *most recent nonempty* value the attribute takes among all profiles of this officer, where *most recent* is defined using the release date of each dataset by the CPD. In this way, if an attribute is empty in some profiles but present in others, a nonempty value will be selected. Choosing the most recent value is justified since (i) it is more likely to still be current (ii) it is more likely to contain the latest corrections made by the CPD to their database. The consolidated profiles for each officer can be found in the file `final/roster.csv`; each row of this file corresponds to a unique officer in the population.

As already alluded to in Section 3.1, the unit assignment data revealed that around 6% of the records display an end date chronologically preceding the start date of the assignment. A closer inspection of these faulty records revealed a systematic pattern: whenever such a record appears, it is possible to find among the other assignments of the same officer another record whose start date is exactly one day after the end date of the faulty record. This led us to formulate the following hypothesis: *the faulty end dates were not manually entered, but were instead automatically generated by the data infrastructure of the CPD*. More specifically, we believe the end dates were added by a piece of computer code which processed all unit assignments in order, and set as the end date of each assignment, the day immediately preceding the start date of the following assignment. The faulty records then arose from the fact that they were wrongly positioned in the order considered by the computer code. We used this hypothesis as the basis for the cleaning of the unit assignment data, whose details are given in Appendix B.3.

Table 2: Counts of officers (first row) and active officers (second row).

| | Gender | | Race | | | | | |
| | M | F | White | Black | Hisp. | Asian/P.I. | Indig. | Bl. Hisp. |
|---|---|---|---|---|---|---|---|---|
| **All** | 28316 | 7122 | 21047 | 8599 | 4811 | 582 | 67 | 9 |
| **Active** | 11118 | 4452 | 7241 | 3895 | 3596 | 467 | 40 | 9 |

# 4 Exploratory Analysis

In this section, we present an exploratory analysis of the different entities present in the cleaned and linked data, with the purpose of providing some insight into the data as well as potential pitfalls in its use. Code for these analyses is available in the `examples/` folder of the repository. Note that throughout this section, officer race and gender are binned per the CPD's coarse categories. Unless otherwise specified, an officer is considered "active" if their resignation date is after 2019-01-01.

**Roster and Units.** Figs. 1 and 2 and Table 2 provide summaries of demographics, age, and unit assignments of the roughly 35 000 officers present in the data, whose appointment dates range from 1936 to 2018. Fig. 1 in particular highlights an important limitation of the data: although there appears to be a steep increase in the number of active officers until the 1980s, it is much more likely that a significant fraction of officers is missing from the database during those early years. Since the process through which officers were added to these records is unclear, the roster data should be assumed to contain only a subset of officers prior to the 1980s. Another point of interest—demonstrated in Fig. 2—is that officers most commonly join precisely 2 units in their career: Unit 44 (the training academy), and their sole assignment.

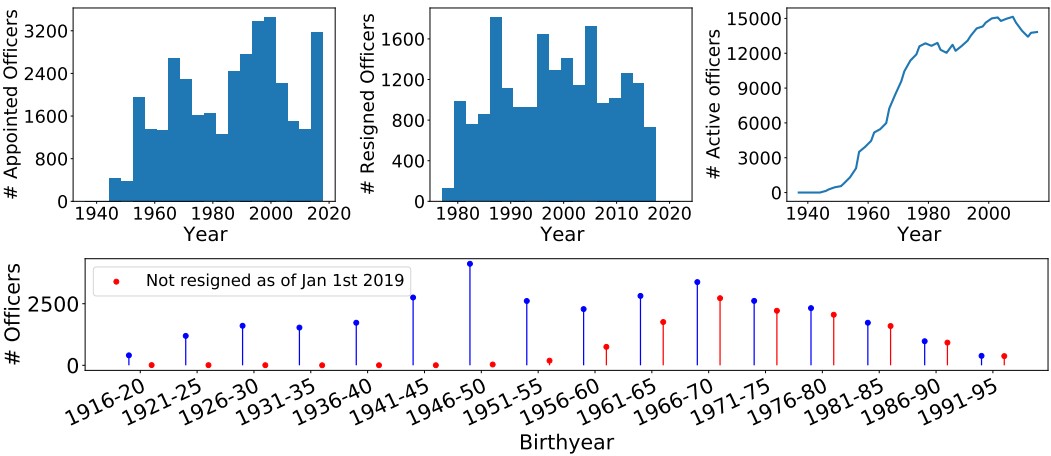

Figure 1: Officer birth years (bottom), appointments (top left), resignations (top center), and active officers (top right) appearing in the CPD roster database from the years 1940 to 2019.

**Complaints.** We report patterns of complaints as functions of both time and officer demographics in Fig. 3. This figure highlights another important feature of the data: the number of complaints filed gradually reduces over the years, potentially as a consequence of the increasing perception of ineffectiveness of such complaints. It is also clear in this figure that male officers of color receive proportionally more complaints than both female officers and white officers, indicating potential racial bias in complaint filings. Supplemental figures in Appendix C further demonstrate that fewer complaints are filed during the weekends and colder months.

**Salary and Awards.** Fig. 4 shows the officer salary for a selection of representative positions versus years of experience, as well as the number of award requests filed for officers by demographic group. Unsurprisingly, salary generally increases with experience and rank, and Fig. 9 in the appendix shows that salary has only a marginal dependency on race and gender, likely because officers are unionized with strict rules about salary progression. However, Fig. 4 and Fig. 8 both hint at implicit forms of

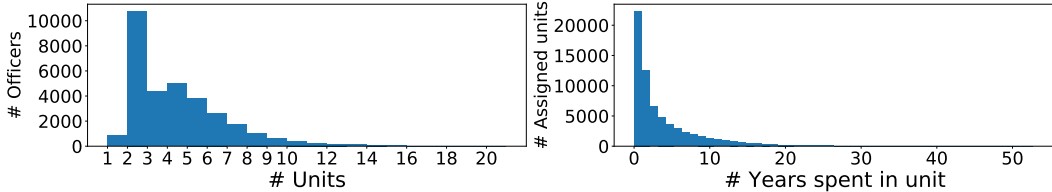

Figure 2: Histograms of the number of unit assignments for officers over their career (left), and the number of years spent in each unit for assignments that had terminated by 2019-01-01 (right).

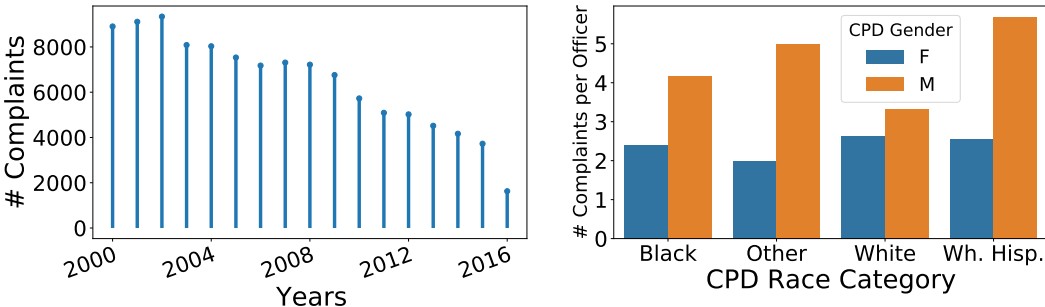

Figure 3: Complaints filed by year (left) and complaints per officer by race and gender (right).

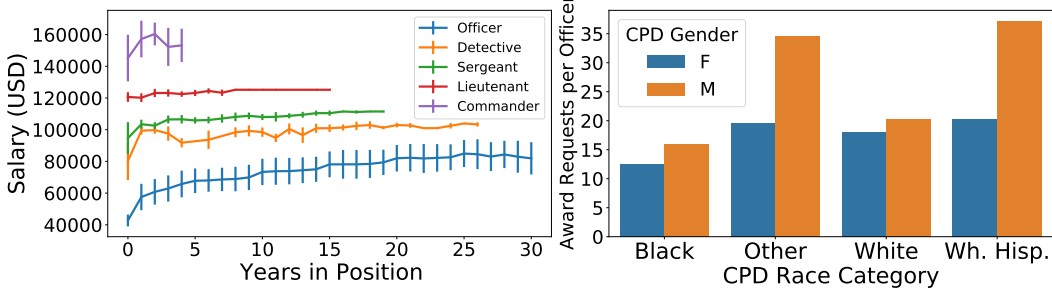

Figure 4: Officer salaries as a function of the number of years in a selection of representative positions (left), and the number of award requests per officer by race and gender (right).

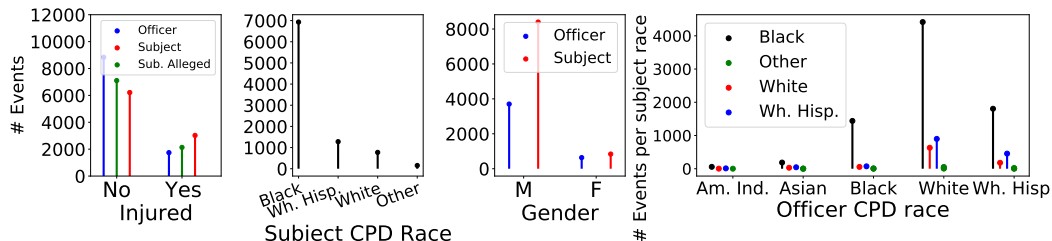

Figure 5: Summary counts of TRRs by injury status, race and gender.

gender bias present in the data. In particular, awards appear to be requested at a higher rate for male officers than for female officers, and similarly male officers disproportionately occupy higher ranks than their female counterparts. Fig. 8 also suggests that white officers are more readily promoted through the lower ranks (officer, detective, sergeant) than officers of color. This trend seems to disappear for higher ranks (lieutenant and above), although drawing firm conclusions would require a rigorous statistical analysis given the much smaller number of officers occupying these ranks.

**Tactical Response Reports.** Fig. 5 presents summaries of the tactical response reports in the data. In most cases, neither the officers nor the subjects involved are injured, although civilians get

injured at a much higher rate (left subplot), and officers tend to fire first (4593 instances versus 329). Analysis involving these data should account for racial bias in officers' use of force: black civilians are the subject of a TRR over 4 times more often than any other race. Figure 6 in the appendix also provides some insight into the temporal nature of TRRs: they are filed more frequently at night, on the weekends, and during warmer months. There is an unusual spike in TRRs between 2010 and 2012 that warrants further investigation.

## 5  Discussion

### 5.1  Intended Uses

Uses for this dataset are varied and rich. For example, researchers could use the data in a wide range of predictive tasks, such as predicting officer misconduct, resignation, and shooting as a function of their underlying demographic data or complaints filed against them. Past work has engaged in this type of analysis on both the raw data underlying the present work, as well as confidential internal police department data [6, 14]. Our dataset can also be used to study social networks (both in the context of policing and more generally). In particular, we can use the complaint data in `complaints_officers.csv` to construct an undirected graph on the set of police officers with an edge between each pair of officers listed together on the same complaint. Moreover, the complaints can be linked to the `tactical_response_reports.csv` file to focus on the subgraph of officers who filed a TRR. We report summary statistics for the corresponding graphs in Table 3, with additional related visualizations in Appendix C. These networks are of interest in and of themselves, but can also be used to investigate the dynamic patterns of officer wrongdoing along such police networks [18]. Existing research has used the complaint data to identify such patterns and to investigate whether pairs of officers connected on a network are more likely to have been accused of misconduct [19]. Finally, this dataset could be used to track the effects of new disciplinary practices, new training techniques, and new oversight on complaints and use of force. Research has explored, for example, whether civilians filed fewer complaints about officers' force in the wake of the Department of Justice investigation of the Chicago Police Department [20].

Table 3: Summary statistics for the complaints network graph, and the subgraph of officers in TRRs. The network is constructed from all complaints and TRRs filed between 2004-01-01 and 2015-12-01. $N$ and $E$ denote the number of nodes and edges respectively. $N_\ell$ and $N_i$ denote respectively the number of nodes in the largest connected component and the number of isolated nodes.

|  | $N$ | $E$ | *Avg. degree* | *Triangles* | *Max clique* | $N_\ell$ | $N_i$ |
|---|---|---|---|---|---|---|---|
| *All* | 14,372 | 106,701 | 14.85 | 361,878 | 64 | 13,950 | 0 |
| *In TRRs* | 4,105 | 22,064 | 10.75 | 44,786 | 28 | 3,822 | 225 |

### 5.2  Ethical Considerations

Recent work on algorithmic fairness focuses on the potential for racially biased data to produce racially biased results [21–23]. This research suggests that race shapes data collection in criminal justice in at least two ways that are likely to affect data collection on black officers. First, dating at least to the 1960s, black officers have been more likely to be assigned to predominantly black neighborhoods and/or to neighborhoods where police interaction is more pervasive [24]. Given an increased frequency of interaction, officers assigned to these neighborhoods may be statistically more likely to be the subject of complaints [25]. Second, owing to cognitive bias, complainants may be more likely to file a complaint against officers of color, either alone or in pairs. Fig. 3 provides initial quantitative evidence towards that point. This racial asymmetry in the collection of complaint data may well produce, for example, racially biased predictions of police misconduct. Relatedly, because our dataset may be used to explore predictive policing of the police, officers of color may be unfairly and disproportionately identified to be at higher risk of misconduct [21–23, 26].

### 5.3  Limitations and Future Work

Using civilian and administrator complaint data to study actual (not merely perceived) police misconduct inevitably faces significant questions about validity. At least one study has found that

because of flaws in police record keeping and categorization, the practice of using complaints to measure police behavior is unreliable [27]. Even so, other research finds a strong correlation between civilian-filed complaints against officers and internal complaints against the same officers filed by other officers or supervisors [28]. Whatever the truth of the matter, the dataset could be strengthened by adding more objective measures of misconduct, for example, data on individual officer misconduct from oversight agencies.

Use of this dataset for network research faces a particular set of limitations. To wit, researchers who use complaint data to generate social networks must acknowledge that the co-listing of two officers on a complaint is an incomplete proxy for a professional network relationship or exposure to another officer's misconduct. Data on partner assignment and dispatches would more accurately reflect officer relationships and their exposure to misconduct.

In general, future work should focus on integrating into this dataset as many objective sources of data on individual officers as possible. Objective data could include information about adverse incident histories, officer discipline histories, counseling interventions, domestic violence incidents, weapons violations, sustained complaints, and lawsuit settlements. Additional information about officer activities could include partner assignments, dispatch information, arrest and stop information, unit leadership, and unit disciplinary history.

## Acknowledgments

T. Campbell was supported by a National Sciences and Engineering Research Council of Canada (NSERC) Discovery Grant and an NSERC Discovery Launch Supplement. Special thanks to Andrew Fan and Rajiv Sinclair of the Invisible Institute for providing pointers to the original data and answering numerous questions.

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

## A  Datasheet

This section provides a datasheet [29] for the dataset repository.

### A.1  Motivation

**For what purpose was the dataset created?**  The original raw data files were sought by J. Kalven, a journalist in the City of Chicago, as part of his investigation into police abuse. After the original FOIA requests and legal case, the non-profit Invisible Institute (`https://invisible.institute`) began to collaborate with Kalven and the University of Chicago's Mandel Legal Aid Clinic to follow up on earlier FOIA requests and to file new ones. The data disclosed in response to these earlier and now ongoing FOIA requests were made available online as part of the Citizens Police Data Project.

**Who created the dataset, and on behalf of which entity?**  The Chicago Police Department (CPD), Civilian Office of Police Accountability (COPA), and the City of Chicago produced the raw data files in response to FOIA requests. The raw data were curated and released publicly by the Invisible Institute and its collaborators. The cleaned and linked data were produced as part of research by the authors of this document.

**Who funded the creation of the dataset?**  The acquisition of the original raw data was funded by the Invisible Institute.

### A.2 Composition

**What do the instances that comprise the dataset represent?**   There are multiple types of instance in this data.

- Officer: information about an individual police officer

- Unit assignment: a single unit assignment for an officer

- Complaint: a complaint filed against a police officer, either internally or by a civilian

- Tactical Response Report: a form that an officer is required to fill out after their response requires use of force

- Award request: a request to grant an award to an officer

- Salary: a record of an officer's salary, pay grade, and position across multiple years

**How many instances are there in total (of each type)?**   There are roughly 35,000 unique officers in the cleaned roster appearing in roughly 130,000 profiles throughout the data, 730,000 award request records, 194,000 salary records, 108,000 unit assignment records, 109,000 complaints, and 10,500 tactical response reports.

**Does the dataset contain all possible instances or is it a sample of instances from a larger set?** This data contains information regarding all sworn officers in the Chicago Police Department / City of Chicago databases for the stated date ranges (which differ for each source of raw data).

**What data does each instance consist of?**

- Officer: officer unique ID, race, gender, age, appointment date, resignation date, badge number(s), position title(s)

- Unit assignment: officer unique ID, start date, end date, unit number

- Complaint: complaint ID, involved officer IDs, allegation, result of the investigation, resulting sanction (where available)

- Tactical Response Report: report ID, event location, date, and time, environmental conditions, who was notified, weapons discharged, weapon information, subject demographic information

- Award Request: awardee unique ID, requester, request date, award reference number, award type, request tracking number, incident dates, ceremony date

- Salary: officer unique ID, salary, position title, pay grade, year

**Is there a label or target associated with each instance?**   Not explicitly. However, labels could be constructed from the data that exists. For example, one could aggregate complaints to produce an integer "number of complaints" for each officer in the data, and use that as the response variable in a prediction task.

**Is any information missing from individual instances?**   In the original raw data files, missing data (of all fields) is quite common (see Appendix D). In the cleaned and linked data files, we are able to aggregate multiple profiles of a single officer appearing throughout the data to "fill in the gaps," although this process is not perfect and there are still missing entries.

**Are relationships between individual instances made explicit?**   In the raw data, no. In the cleaned data, we provide a unique officer identification that enables linking the activities and records regarding individual officers across datasets. There is no relational data (i.e., network edges) explicitly contained in the data. However, it is possible to use the data to construct a network, e.g., by linking officers co-listed on complaints.

**Are there recommended data splits?**   No, although the officer database is likely to be incomplete prior to roughly 1980.

**Are there any errors, sources of noise, or redundancies in the dataset?** There are redundancies in the raw data, but these are removed by our cleaning and linking procedure. Errors, inconsistencies, and missing data are also present in the raw data; our cleaning and linking resolves much of these issues. However, per Section 4, the officer database is likely to be incomplete prior to roughly 1980 (as officers were added to the database only gradually over time).

**Is the dataset self-contained, or does it rely on external resources?** The dataset is self-contained: the raw data itself is stored in the `raw/` folder of the repository (with links to the external source files for reference), and the cleaned/linked data is produced by the source code in the repository.

**Does the dataset contain data that might be considered confidential?** No; all of this data was publicly released as part of FOIA requests. Confidential data (e.g., relating to under cover officers) was withheld by the Chicago Police Department.

**Does the dataset contain data that, if viewed directly, might be offensive, insulting, or threatening?** No.

**Does the dataset relate to people?** Yes; it contains records relating to police officers in the Chicago Police Department.

**Does the dataset identify any subpopulations?** Yes; officer records include race, gender, age, appointment date, unit history, badge numbers, position title, salary, awards, complaints, and tactical response reports. Subpopulations of officers can be constructed using these fields.

**Is it possible to identify individuals?** Yes; detailed information is available that could be used to identify individual officers.

**Does the dataset contain data that might be considered sensitive in any way?** The data contains a coarse categorization of racial origins of officers.

## A.3 Collection Process

**How was the data associated with each instance acquired?** The raw data were obtained via FOIA requests to the City of Chicago and Chicago Police Department.

**What mechanisms or procedures were used to collect the data?** The raw data were obtained via FOIA requests to the City of Chicago and Chicago Police Department.

**If the data are a sample from a larger set, what was the sampling strategy?** Not applicable.

**Who was involved in the data collection process and how were they compensated?** Journalists in collaboration with the Invisible Institute were responsible for filing the FOIA requests, and officials within the Chicago Police Department and City of Chicago were responsible for providing data in response to those requests. It is not known explicitly whether or how either party was compensated.

**Over what timeframe was the data collected?** The earliest releases per FOIA request occurred in 2016, and continue to occur as more FOIA requests are filed. The raw data itself pertain to records from the CPD dating back to the mid 20th century. The roster data covers the period up to 2018. The awards data pertains to records from 1967 to 2019. The salary data pertains to the years 2002 to 2017. The unit history data covers records up to 2016. The complaints data pertains to records from 1967 to 2016. The tactical response report data pertains to records from 2004 to 2017.

**Were any ethical review processes conducted?** It is unknown whether the CPD conducted any ethical review processes prior to the release of the raw data. No ethical review process was conducted prior to the activities involved in the present repository, i.e., cleaning the publicly available data.

**Does the dataset relate to people?** Yes; it contains detailed records regarding the activities of police officers in the City of Chicago.

**Did you collect the data from the individuals directly, or obtain it via third parties?** The raw data was acquired from public links provided by the Invisible Institute (`https://invisible.institute`). The Invisible Institute acquired the data through FOIA requests made to the CPD and the City of Chicago.

**Were the individuals notified about the data collection?** It is unknown whether the individual officers were notified by the CPD when the raw data was released.

**Did the individuals in question consent to the collection and use of their data?** Not explicitly. The Chicago Police Department was compelled by law to produce these records per FOIA requests.

**If consent was obtained, were the consenting individuals provided with a mechanism to revoke their consent in the future or for certain uses?** Not applicable.

**Has analysis of the potential impact of the dataset and its use on data subjects been conducted?** Not known.

### A.4 Preprocessing and cleaning

**Was any preprocessing of the data done?** Yes; the main section of this documentation provides details the cleaning and linking of the raw data resulting from FOIA requests made to the City of Chicago.

**Was the "raw" data saved in addition to the cleaned data?** Yes; the raw data is available in the `raw/` folder in the repository.

**Is the software used to clean the data available?** Yes; the source for cleaning and linking is provided in the `src/` folder in the repository.

### A.5 Uses

**Has the dataset been used for any tasks already?** Not the newly cleaned and linked version. The raw data itself has been used previously; see Section 5 for details.

**Is there a repository that links to any or all papers that use the dataset?** Not that the authors of this work are aware of.

**What (other) tasks could the dataset be used for?** This data set has a rich variety of possible uses; for example, network analysis (and in particular, analysis of dynamic events occurring on networks) and predictive regression/classification. See Section 5 for more details.

**Is there anything about the composition of the dataset or the way it was collected and cleaned that might impact future uses?** Yes; the data are less reliable in earlier years (e.g., pre-1980). See Section 4 for more details.

**Are there tasks for which the dataset should not be used?** This data should not be used to single out, study, or identify individual officers.

### A.6 Distribution

**Will the dataset be distributed to third parties outside of the entity on behalf of which the dataset was created?** Yes, the data is publicly available.

**How will the dataset be distributed?** It is available on GitHub at `https://github.com/chicago-police-violence/data`. Release versions will be marked using the "release" feature on GitHub.

**When will the dataset be distributed?** It is currently publicly accessible.

**Will the dataset be distributed under a copyright, other IP license, or terms of use?** Yes; the source code is released under the MIT license, and the data output by the cleaning code is released under the Creative Commons 4.0 BY-NC-SA license.

**Have any third parties imposed IP-based or other restrictions on the data associated with the instances?** No.

**Do any export controls or other regulatory restrictions apply to the data?** No.

### A.7    Maintenance

**Who is supporting/hosting/maintaining the dataset?** The repository will be hosted on GitHub. As of August 2021, the repository owners are Thibaut Horel, Trevor Campbell, and Lorenzo Masoero, but ownership may change over time.

**How can the data owner/curator be contacted?** Issue threads on GitHub are the primary channel of contact for the repository maintainers.

**Is there an erratum?** Not as of yet. For each major release version, notes will be included and hosted in the repository that will detail cleaning/linking errors that have been fixed.

**Will the dataset be updated?** The original raw source data from FOIA requests will not be modified. More raw data files may be added over time corresponding to new FOIA requests. The data cleaning and linking code will be edited over time to fix errors; release versions will be clearly marked on GitHub. There is no set schedule for updates.

**If the dataset relates to people, are there applicable limits on the retention of data associated with the instances?** No; this data was released per FOIA requests and is in the public domain.

**Will older versions of the dataset continue to be supported/hosted/maintained?** Yes; a full version-controlled history of the project exists on GitHub.

**If others want to extend/augment/build on/contribute to the dataset, is there a mechanism for them to do so?** Yes; the repository for the dataset is hosted on GitHub, where pull requests are a usual channel for external contribution.

