# OpenReview forum: "The CPD Data Set: Personnel, Use of Force, and Complaints in the Chicago Police Department"
_NeurIPS.cc/2021/Track/Datasets_and_Benchmarks/Round2 — NeurIPS 2021 Datasets and Benchmarks Track (Round 2)_

### Official Review · Reviewer_WAob · 2021-09-20
**Describes an approach and provides code for reliably cleaning and linking various FOIA-requested data about the Chicago Police Department.**

**Rating:** 7
**Confidence:** 3

**Strengths:**

- Well written paper and organized code repository.
- The work described is already a great contribution to the research community and has a high likelihood of creating positive impact. Open and reproducible approaches to cleaning and linking data are very valuable. The effort to do so with a high level of transparency (e.g. open code, open GitHub issues outlining future directions) and many worked examples (e.g. Jupyter notebook files) will be helpful.
- The general approach can be useful for other contexts, especially for linking various documents and releases obtained via FOIA or similar procedures. It may even be possible that some of the software and methods choices are directly re-usable.
- The topic is of high societal importance.
- Good overview of challenges with this kind of data.

**Weaknesses:**

- Some of the insights gleaned from exploratory analysis seemed brief and weakly supported. More discussion of this point in "Correctness" below.
- References to CPD data infrastructure were a bit unclear (e.g. "probably due to internal errors in the CPD data infrastructure", ) Is there any way to confirm the "data infrastructure bug" hypothesis in the future (e.g. a new FOIA request aimed specifically at answering this question?), or is this likely the best we can do?
- Cursory connection to journalistic coverage on the topic, which is a highly relevant area of related work. For instance, is there any corroborating evidence for decline in complaints or an explanation for why this happened? Some specific examples below in "Relation to Prior Work"
- Discussion of negative impacts could be expanded, especially regarding the extent to which many of the problems with using police-generated data for ML will persist for using this data. Some specific examples below in "Ethics".

**Additional Feedback:**

Very minor comment: the Jupyter notebooks in current version of the repo are "un-run" versions. The authors may want to consider generating some examples as HTML or PDF for some interested users.

I was also curious if there was a strong reason for not providing the `final` data files directly in the repo (e.g., to make sure users have the most up-to-date version?). This may be useful for some dataset users.

**Clarity:**

Organization of the paper is very clear and the contribution is communicated well. No concerns here.

**Correctness:**

Data cleaning choices seems uncontroversial. Procedure to build a database of officer UIDs seems correct, although it may help to further describe any manual validation procedures (e.g. did authors look at a sample of merged entities?).

The procedure for generating "canonical profiles" of officers seems reasonable and is explained well.
Documentation of each FOIA data file is important for researchers seeking to understand the history of the data.

The conclusions of the exploratory analysis were introduced very briefly. Overall, it was not clear if the data as provided was enough to support these conclusions (e.g. the claims about why complaint volume has fallen, the explanation for disparities in complaints by officer demographics). It may strengthen the paper to be make more specific suggestions around what external data sources or new analysis techniques are needed to answer these questions (e.g. more FOIA requests? Scraping data? Using causal inference techniques or quantitative fairness techniques? etc.)

**Documentation:**

The rationale behind cleaning and linking is Section 3 is well-documented. Accompanying code is a useful supplement for understanding specific details.

Section 4 provides an exploratory analysis. Code examples are also provided in separate notebook files, which is likely very helpful for interested readers.

Hosting (via GitHub) and licensing (CC-BY-NC-SA) seem reasonable.

**Ethics:**

The paper rightfully notes the potential for "racially biased data to produce racially biased results". While this has been highlighted in criminal justice contexts, the paper discusses how this may apply to police officers themselves.

However, the paper does not discuss the extent to which training predictive models using data provided by police departments can serve to rationalize certain data uses (see Section 3.3. of [3] for more on this concern). Concretely, I think taking a firmer stance on which particular predictive tasks should be treated as legitimate will strengthen the contribution of the paper. For researchers who accept the argument that predicting the likelihood someone will engage in criminal activity (i.e. predictive policing [4]) is problematic, is predicting officer misconduct also problematic? The "predictive" intended uses seem quite different from the "evaluate effects of a new policy" type of uses.


[3] Paullada, A., Raji, I. D., Bender, E. M., Denton, E., & Hanna, A. (2020). Data and its (dis) contents: A survey of dataset development and use in machine learning research. _arXiv preprint arXiv:2012.05345_.


[4] Heaven, 2020. Predictive policing algorithms are racist. They need to be dismantled. https://www.technologyreview.com/2020/07/17/1005396/predictive-policing-algorithms-racist-dismantled-machine-learning-bias-criminal-justice/

**Relation To Prior Work:**

Quick summary of work drawing on CPD FOIA files is provided (specifically, refs 12,13, and 14 from the paper). It may be useful to additionally specify what kind of findings this work produced, i.e. to contrast and compare with the intended uses of this dataset.

Regarding connections to journalistic coverage of police complaints, etc., articles along these lines could be relevant to discuss in the body of the paper. Or, if the authors think this is out of scope for this particular paper, explicitly mentioning this kind of work could be a helpful lead for users of the dataset.
- This article from the Chicago Tribune [1] mentions that "Nearly 60 percent of all the complaints were thrown out without being fully investigated because the alleged victims failed to sign required affidavits."
- Another Tribune article [2] mentions that "a Tribune analysis of police complaint data showed about 60% of approximately 17,700 complaints over a four-year period ending in mid-December 2014 were thrown out because there were no signed affidavits."


[1] Gorner and Hing, 2015. Tribune analysis: Cops who pile up complaints routinely escape discipline https://www.chicagotribune.com/news/ct-chicago-police-citizen-complaints-met-20150613-story.html


[2] Gorner, 2021. Filing a complaint against a Chicago cop? You wouldn’t have to sign your name to it under proposed law https://www.chicagotribune.com/news/criminal-justice/ct-chicago-police-affidavits-anonymous-complaints-20210129-str7gjrkkzaf3bqypclw7na3t4-story.html

**Summary And Contributions:**

This paper addresses a lack of open and reproducible software and data for investigating police activity. The authors describe a method for cleaning and linking Freedom of Information Act (FOIA)-acquired data about the Chicago Police Department. They provide code to easily perform this cleaning and linking task. The paper provides brief exploratory analyses and discusses intended uses of the dataset and some ethical concerns with predictive work in this domain.

---

### Official Review · Reviewer_Q5EA · 2021-09-20
**Important work that allows researchers to study police misconduct**

**Rating:** 8
**Confidence:** 3

**Strengths:**

- The authors provide data on police’s use of force and possible complaints – this is the opposite of the typical approach which is to collect data about the people interacting with police.
- The authors did a great job in preprocessing the data and documenting the process.
- I really appreciate the datasheet in Appendix A. I was wondering if that could be provided in a more prominent place in the repository – perhaps as its own .md file at the top level of the repo?

**Weaknesses:**

- The original repository by Invisible Institute doesn’t seem to be particularly active at the moment. Why did the researchers feel compelled to create a new repository instead of contributing to the existing one?
- Could the authors say something about to what extent we can expect findings from this dataset to generalize? For example, does this data generalize to other cities / states / countries? What’s different about the CPD that would limit generalizations?


**Additional Feedback:**

- Line 211 f.: “and set as the end date of each assignment, the day immediately preceding the start date of the following assignment” --> the comma after “assignment” seems to be misplaced

**Clarity:**

Overall, the paper is written in a very clear way. There were just some smaller things that would be helpful to clarify:

- Line 67 ff.: “Furthermore, the Department of Justice (DOJ) only began last year to collect data on arrest-related deaths through the implementation of the “Death in Custody Reporting Act” of 2013 [10].” --> It is unclear to me why the collection of data only started last year if the Act is from 2013 – is this because of delays in the implementation of the Act?
- Line 220 f.: “Unless otherwise specified, an officer is considered “active” if their resignation date is after 2019-01-01.” --> I don’t understand why officers who have resigned are considered active. Could you provide your reasoning for that?
- Figure 1, bottom: This figure is unclear to me. Only the red dot appears in the legend, what do the blue dots represent? And do the red dots show us that for officers born before 1950, the resignation date is basically always missing? The insight from this figure could be stated in the text in 1 or 2 sentences.
- For people unfamiliar with (American) law enforcement: What are awards and when are they typically requested?

**Correctness:**

The dataset has been constructed in a sound way from the existing raw data. The authors clearly highlight limitations of the dataset, such as missing data and difficulties with linking the dataset.

**Documentation:**

- Documentation: Very well-documented dataset. The authors make understanding the repository as easy as possible. One caveat: I noticed that the descriptions of some columns are missing in the description.md (although they seem pretty self-explanatory, so this is not strictly necessary).
- Intended uses: Yes, examples are given. Would be good to add what *not* to do with the data. This is mentioned in the datasheet in Appendix A, but it might be helpful to mention this in the main text, too.
- URL: Yes
- License: Yes, MIT license
- Maintenance plan: The maintenance plan is part of the datasheet, but it might be useful to have a Contribution.md in case people want to add more data collected through FOIA requests. In particular, I am wondering how the authors plan to keep track of changes in the original repo (created by Invisible Institute)? Invisible Institute’s repo doesn’t seem to be active at the moment, but in the case of changes, do they plan to include them in their repo? In the case that the authors acquire more raw data, do they plan to also add that to Invisible Institute’s repository? Or are there plans to talk to Invisible Institute to merge the two repositories together?


**Ethics:**

Ethical concerns are described in Section 5.2.

Two additional remarks / questions on that:

- How should the bias reported on in Section 5.2 be dealt with when working with the data to, e.g., predict excessive use of force? Should such bias, for example, be corrected for? Should predictive tools be evaluated with respect to their fairness (e.g., through fairness metrics)?
- One of the letters (https://github.com/chicago-police-violence/data/blob/main/raw/P4-41436/P441436_-_Letter.pdf) states that the names of undercover officers have been redacted. Is there a chance that the names of these officers still appear in some of the other datasets in case that the data was released before their time as undercover officers? And is there a chance that the pairwise matching reidentifies these officers? Or falsely identifies some officer as an undercover officer?


**Relation To Prior Work:**

Related work is appropriately discussed. One work to potentially bring up is “Studying Up: Reorienting the study of algorithmic fairness around issues of power” by Barabas et al. as the CPD dataset allows for what Barabas et al. call “studying up,” i.e., studying people and institutions with power, such as the police.

**Summary And Contributions:**

The researchers provide a cleaned and pre-processed dataset about individuals in law enforcement. This is a novelty because data about police interactions so far has focused on just one side: the members of the public that have interacted with the police – but not the police itself. Because of this skewed access to data, it is incredibly difficult to study police and usually only indirectly possible. While the data that the paper publishes is not new (it has been collected, used and published by Invisible Institute), the paper makes the data easier to use for other researchers. This paper therefore makes a significant contribution in providing clean and easy-to-use data on police officers.

---

### Official Review · Reviewer_WkMp · 2021-09-21
**Valuable dataset and processing code to support analyses of critical social issue in an important geography**

**Rating:** 8
**Confidence:** 3

**Strengths:**

This dataset – which takes vasts amounts of data that should be made available to the public but is often inaccessible due to the significant amounts of time needed for both FOIA requests and preparation/cleaning – could enable a variety of new empirical research on patterns of law enforcement as well as the potential impact of new interventions that has prior to its launch not been available to the public. This dataset is the culmination of several years of perseverance and commitment to unveiling what is often obfuscated.

**Weaknesses:**

The dataset on this important city is very relevant. If only we could have access to similar datasets across the country!


**Additional Feedback:**

Bravo! This is an exceptional dataset and cleaning protocol which has been built on many years of perseverance and teamwork.



**Clarity:**

The paper is well written and clear. The graphics are also helpful and well-explained.

**Correctness:**

The dataset appears to be constructed soundly and is well documented. Where assumptions were needed in order to proceed with processing, those are clearly indicated in the dataset. Observed limitations are made explicit.

**Documentation:**

Original raw data is provided, processed data with all processing steps provided, github repo maintains the files that outline in explicit detail how the data were munged, and an organization exists that hosts and will presumably also maintain the dataset. The appendix includes a useful datasheet for datasets that also enables easier review of it.


**Ethics:**

These data and the clean analysis codes are of such importance to empirical investigations to policy activity that if this data and the corresponding code is *not* able to be shared, then we encounter some ethical problems.

**Relation To Prior Work:**

The manuscript reflects a clear linkage to the existing problems in the literature and a knowledge of what their dataset has the potential to enable as distinct from other difficult to access datasets that are also limited in scope.

**Summary And Contributions:**

This dataset contains information on the personnel, activities, use of force, and complaints in from over ~35,000 police officers in the Chicago Police Department spanning over a decade. Prior to the authors’ extensive and well-documented procedures for cleaning of this data obtained from several FOIA requests, the data was unlinked, inconsistent, and furthermore undocumented. This dataset – which takes vasts amounts of data that should be made available to the public but is often inaccessible due to the significant amounts of time needed for both FOIA requests and preparation/cleaning – could enable a variety of new empirical research on patterns of law enforcement as well as the potential impact of new interventions that has prior to its launch not been available to the public. The dataset *and* the documentation surrounding how it’s processed – centers on a important and relevant location in the US where investigations of policy activity are common but empirical evaluation is thin – in part due to the prior inaccessibility of sufficient data. The authors outline several important possible strands of research that this important and extensive dataset enables, ranging from prediction to causal inference and beyond.

---

### Comment · Program_Chairs · 2021-10-13
**Official Ethics Review**

We acknowledge the ethical issues raised in section 5.2 and the demographic reporting in Figure 5. Since authors use public information for accountability purposes, consent and privacy concerns seem to be minimized, and authors had furthermore reported passing IRB requirements.

Although it seems that authors focus on the declared intended use highlighted in 5.1, it is possible for others to imagine potentially harmful use cases for this dataset, including using the information for targeted harassment or violence against specific police officers. Since the dataset could potentially reveal police identities, authors should think seriously about how to restrict and communicate throughout the data distribution process. Proposals such as a data license [1] may be a useful framework for thinking about how to approach this task, in addition to setting terms of use or requiring explicit requests for access.

[1] Benjamin, Misha, et al. "Towards standardization of data licenses: The montreal data license." arXiv preprint arXiv:1903.12262 (2019).

---

> ### Author Response · Authors · 2021-10-14
> **Thank you for your review**
>
> We are very grateful for the suggestion to use a license specifically tailored for data, as opposed to a more “generic” Creative Commons license. We are currently looking into the suitability of the Montreal Data License for our release.
>
> In our view, this project, which consists of cleaning and organizing publicly available data, does not increase the risk of revealing police identities beyond the existing risk from the publicly available datasets. The public data are readily downloadable from the [Invisible Institute's website](https://invisible.institute/download-the-data) and [Citizens Police Data Project](https://cpdp.co/). The identities of police officers employed by the Chicago Police Department have already been revealed by these previous releases (following FOIA requests). Indeed, the [search](https://cpdp.co/search/) page from the Citizens Police Data Project allows searching for a police officer by name much more efficiently than would be possible with our dataset (which is not indexed on officer name). In contrast to the Citizens Police Data Project, it would take a non-trivial amount of work to recover an officer’s name from our dataset.
>
> As a result, enforcing a restriction on the use of our dataset to prevent harassment against individual officers would be difficult if not impossible in light of the already existing data and tools; it would be difficult even to assess whether the transgressor used our dataset or the existing public data. Note that our dataset does not identify any additional officers, and in particular, does not reveal those redacted from the FOIA responses.

---

### Decision · Program_Chairs · 2021-10-09

**Decision:**

Accept

**Comment:**

In this paper the authors provide a new dataset, based on Freedom of Information Act requests, covering complaints against the Chicago Police Department. The authors detail how they process and clean the data and present an exploratory analysis based on the data. All reviewers found the paper to be a valuable contribution to the field. That said, given the sensitivity of this area (and open challenges raised by multiple reviewers), in my opinion acceptance is conditional on an ethics review.

Flagged for an additional ethics review because of data access and privacy issues.